# Novel Yellow Azo Pyridone Derivatives with Different Halide Atoms for Image-Sensor Color Filters

**DOI:** 10.3390/molecules27196601

**Published:** 2022-10-05

**Authors:** Sunwoo Park, Yuna Kang, Hyukmin Kwon, Hayeon Kim, Seokwoo Kang, Hayoon Lee, Chun Yoon, Jongwook Park

**Affiliations:** 1Integrated Engineering, Department of Chemical Engineering, Kyung Hee University, Yongin 17104, Korea; 2Department of Chemistry, Sejong University, 98 Gunja-dong, Gwangjin-gu, Seoul 143-747, Korea

**Keywords:** azo pyridine, yellow colorant, dye, image sensor, color filter

## Abstract

Novel yellow azo pyridone dye derivatives were synthesized for use in image-sensor color filters. The synthesized compounds have a basic chemical structure composed of azo, hydroxy, amide, and nitrile groups as well as different halide groups. New materials were evaluated on the basis of their optical, thermal, and surface properties under conditions mimicking those of a commercial device fabrication process. A comparison of their related performance revealed that, among the four prepared compounds, 5-((4,6-dichlorocyclohexa-2,4-dien-1-yl)diazenyl)-6-hydroxy-1,4-dimethyl-2-oxo-1,2-dihydropyridine-3-carbonitrile (Cl-PAMOPC) exhibited the best performance as an image-sensor color filter material, including a solubility greater than 0.1 wt% in propylene glycol monomethyl ether acetate solvent, a high decomposition temperature of 263 °C, and stable color difference values of 4.93 and 3.88 after a thermal treatment and a solvent-resistance test, respectively. The results suggest that Cl-PAMOPC can be used as a green dye additive in an image-sensor colorant.

## 1. Introduction

Recently, color filters have become critical in various electronic devices. In particular, they are used in liquid crystal displays, organic light-emitting diodes, and image sensors such as digital cameras and cameras incorporated into smart phones and tablet PCs. Image sensors can be classified as charge-coupled device (CCD) types and complementary metal-oxide-semiconductor (CMOS) types. In the past, CCD image sensors led the market; however, the market for CMOS image sensors is now increasing because of their advantages of lower cost, lower power consumption, and easier integration than CCDs. The color-filter technology used in image sensors requires finer pixel sizes and higher contrast ratios to achieve higher-quality real images [1]. The pigment dispersion method is most widely used to make red, green, and blue color filters because of the high thermal-, chemical-, and photo-stability of pigments. However, this method is limited by the low solubility of pigments, which makes producing smaller particles difficult and leads to light scattering by aggregated particles [1,2,3,4]. To resolve these problems, dye-based color filters have been studied [5,6,7,8]. Because dyes are soluble in solvents, light scattering by particles is not observed, and light transmittance is therefore improved. In addition, improved thermal and solvent stability in industrial solvents such as propylene glycol monomethyl ether acetate (PGMEA) must be realized for these dyes to be used in dye-based color filters [2,5,6,7,8]. If dyes are not stable in common industrial solvents such as PGMEA, then their application in color filters will be difficult [9,10,11]. Therefore, it is necessary to develop dyes with good thermal and solvent stability [10,11].

In the present study, we synthesized novel yellow dyes based on azo pyridone derivatives bearing methyl or halogen groups and investigated their suitability for use in green color filters. Green colorants cannot perfectly absorb blue light, and yellow additives should be added to a green colorant mixture to achieve high color purity of green [7]. Several dyes have been reported for yellow color, and a popular commercial yellow pigment is Y138 (Figure 1) [12,13,14,15]. The synthesized azo pyridone derivatives were studied to optimize their optical, thermal, and solvent stabilities depending on the chemical structures, and their potential as yellow color-filter materials was evaluated.

## 2. Experimental

### 2.1. Materials and Instrumentation

All reagents used in these experiments were purchased from Sigma-Aldrich or Tokyo Chemical Industry (TCI); their purity was 98% or higher, and they were used without further purification. Y138-millbase was purchased from SKC hi-tech and marketing. ^1^H and ^13^C nuclear magnetic resonance (NMR) spectra were recorded on a JNM-ECZ400S/L1 spectrometer, and high-performance liquid chromatography (HPLC) was carried out on a Shimadzu Nexera UHPLC. Ultraviolet–visible (UV–Vis) optical absorption spectra were recorded using a Lambda 1050 UV–Vis spectrophotometer (Perkin Elmer). Thermogravimetric analysis (TGA) was conducted using a TA Instruments Q5000 IR/SDT Q600 with the sample under an air atmosphere. Radical onset temperature (T_onset_) and maximum temperature (T_peak_) of the compounds were measured by differential scanning calorimetry (DSC), Discovery DSC25 (TA instruments), under the nitrogen atmosphere. Atomic force microscopy (AFM) images were acquired using a Bruker Dimension 3100 SPM. The color difference, ΔE_ab_, was measured using an Otsuka Electronics MCPD-3000 array spectrometer.

### 2.2. Synthesis and Characterization of the Synthesized Azo Pyridone Derivatives

#### 2.2.1. Synthesis of 6-Hydroxy-1,4-dimethyl-2-oxo-1,2-dihydropyridine-3-carbonitrile

Ethyl acetoacetate 6.5 g (50 mmol), methylamine (1 eq), ethyl cyanoacetate (1.1 eq), and piperidine (0.35 eq) were added to 70 mL of ethanol; the temperature was then raised to 78 °C, and the mixture was stirred for 8 h. After completion of the reaction, the mixture was cooled to room temperature. The solvent was removed under vacuum, and then hydrochloric acid (37%) was slowly added to the mixture in an ice bath, followed by stirring for 2 h. The reaction solution was slowly added to ethanol to induce precipitation, and the product was washed with hexane (5.98 g, yield 73%). ^1^H NMR (300 MHz, DMSO-d_6_) δ 5.59 (s, 1H), 3.25 (s, 3H), and 2.20 (s, 3H).

#### 2.2.2. Synthesis of 5-((2,4-Dimethylphenyl)diazenyl)-6-hydroxy-1,4-dimethyl-2-oxo-1,2-dihydropyridine-3-carbonitrile (M-PAMOPC)

2,4-Dimethyl aniline (6.1 g, 1 eq) was added to 16.4 mL of hydrochloric acid, and the resultant solution was cooled to 0 °C. Then, 25 mL of deionized water (DI water) and sodium nitrite (1.02 eq) were added, and the mixture was stirred for 45 min. Sulfamic acid (0.02 eq) and 150 mL of DI water:acetone (1:1) were added sequentially, followed by the slow addition of 6-hydroxy-1,4-dimethyl-2-oxo-1,2-dihydropyridine-3-carbonitrile (1 eq); the resultant mixture was stirred in an ice bath for 5 h. The solid was collected by filtration under reduced pressure and washed with DI water. A yellow dye was obtained by recrystallization from acetone. Reddish brown powder (6.11g, yield 41%).^1^H NMR (400 MHz, CDCl_3_) δ 7.65 (d, J = 8.3 Hz, 1H), 7.13 (d, J = 8.3 Hz, 1H), 7.07 (s, 1H), 3.37 (s, 3H), 2.63 (s, 3H), 2.44 (s, 3H), 2.35 (s, 3H). ^13^C NMR (101 MHz, CDCl_3_) δ 162.09, 160.47, 158.66, 137.96, 136.87, 132.09, 128.68, 126.98, 123.01, 115.82, 114.76, 100.83, 77.43, 77.32, 77.11, 76.79, 26.43, 21.24, 17.04, 16.66, 0.09. High-resolution mass spectrometry (HRMS) (Fast Atom bombardment mass spectrometry (FAB-MS), *m*/*z*): calcd. for C16H16N4O2, 296.33; found: 296.12 [M]+. FT-IR (cm^−1^): 2918 (OH), 2218 (C≡N), 1643–1342 (C=C) 1678 (C=O), 1582 (N=N), 1261 (C-N), 1088–650 (C-H). Purity (high-performance liquid chromatography, HPLC): 94.1%.

#### 2.2.3. Synthesis of 5-((2,4-Difluorophenyl)diazenyl)-6-hydroxy-1,4-dimethyl-2-oxo-1,2-dihydropyridine-3-carbonitrile (F-PAMOPC)

2,4-Difluoroaniline (6.5 g, 1 eq) was added to 16.4 mL of hydrochloric acid, and the resultant solution was cooled to 0 °C. Then, 25 mL of DI water and sodium nitrite (1.02 eq) were added, and the mixture was stirred for 45 min. Sulfamic acid (0.02 eq) and 150 mL of DI water:acetone (1:1) were then added sequentially, followed by the slow addition of 6-hydroxy-1,4-dimethyl-2-oxo-1,2-dihydropyridine-3-carbonitrile (1 eq); the resultant mixture was stirred in an ice bath for 5 h. The solid was collected by filtration under reduced pressure and washed with DI water. A yellow dye was obtained by recrystallization from acetone. Brown powder (5.66 g, yield 37%). ^1^H NMR (400 MHz, CDCl_3_) δ 7.80–7.70 (m, 1H), 7.05–6.96 (m, 2H), 3.37 (s, 3H), 2.61 (s, 3H). ^13^C NMR (101 MHz, CDCl_3_) δ 161.85, 160.07, 158.27, 126.16, 124.27, 117.94, 117.85, 114.23, 113.17, 113.13, 112.94, 112.90, 105.26, 105.04, 104.99, 104.77, 102.76, 77.44, 77.32, 77.12, 76.80, 26.52, 16.65, 0.08. HRMS (FAB-MS), *m*/*z*): calcd. For C14H10F2N4O2, 304.26; found: 304.07 [M]+. FT-IR (cm^−1^): 2976 (OH), 2230 (C≡N), 1744 (C=O), 1702–1305 (C=C), 1587 (N=N), 1240 (C-N), 1102 (C-F) 950–500 (C-H). Purity (HPLC): 97.1%. Melting temperature: 138 °C.

#### 2.2.4. Synthesis of 5-((4,6-Dichlorocyclohexa-2,4-dien-1-yl)diazenyl)-6-hydroxy-1,4-dimethyl-2-oxo-1,2-dihydropyridine-3-carbonitrile (Cl-PAMOPC)

2,4-Dichloroaniline (8.1 g, 1 eq) was added to 16.4 mL of hydrochloric acid, and the resultant mixture was cooled to 0 °C. Then, 25 mL of DI water and sodium nitrite (1.02 eq) were added, and the mixture was stirred for 45 min. Sulfamic acid (0.02 eq) and 150 mL of DI water:acetone (1:1) were then added sequentially, followed by the slow addition of 6-hydroxy-1,4-dimethyl-2-oxo-1,2-dihydropyridine-3-carbonitrile (1 eq); the resultant mixture was stirred in an ice bath for 5 h. The solid was filtered under reduced pressure and washed with DI water. A yellow dye was obtained by recrystallization from acetone. Orange red powder (5.07 g, yield 40%).^1^H NMR (400 MHz, CDCl_3_) δ 7.74 (d, J = 8.9 Hz, 1H), 7.48 (dd, J = 2.2, 0.9 Hz, 1H), 7.36 (ddd, J = 8.8, 2.3, 0.8 Hz, 1H), 3.38 (s, 3H), 2.62 (s, 3H). ^13^C NMR (101 MHz, CDCl_3_) δ 161.66, 159.98, 158.19, 136.57, 132.53, 129.98, 128.93, 124.65, 123.60, 117.70, 114.11, 103.40, 77.43, 77.32, 77.12, 76.80, 26.61, 16.72, 0.09. HRMS (FAB-MS), *m*/*z*): calcd. for C14H10Cl2N4O2, 337.16; found: 336.01 [M]+. FT-IR (cm^−1^): 2972 (OH), 2232 (C≡N), 1733 (C=O), 1701–1307 (C=C), 1585 (N=N), 1215 (C-N), 1045 (C-Cl), 903–539(C-H). Purity (HPLC): 96.6%. Melting temperature: 180 °C.

#### 2.2.5. Synthesis of 5-((4,6-Dibromocyclohexa-2,4-dien-1-yl)diazenyl)-6-hydroxy-1,4-dimethyl-2-oxo-1,2-dihydropyridine-3-carbonitrile (Br-PAMOPC)

2,4-Dibromoaniline (12.5 g, 1 eq) was added to 16.4 mL of hydrochloric acid, and the resultant mixture was cooled to 0 °C. Then, 25 mL of DI water and sodium nitrite (1.02 eq) were added and the mixture was stirred for 45 min. Sulfamic acid (0.02 eq) and 150 mL of DI water:acetone (1:1) were then added sequentially, followed by the slow addition of 6-hydroxy-1,4-dimethyl-2-oxo-1,2-dihydropyridine-3-carbonitrile (1 eq); the resultant mixture was stirred in an ice bath for 5 h. The solid was collected by filtration under reduced pressure and washed with DI water. A yellow dye was obtained by recrystallization from acetone. Dark yellow powder (12.5 g, yield 59%). ^1^H NMR (400 MHz, CDCl_3_) δ 7.78 (d, J = 2.1 Hz, 1H), 7.65 (d, J = 8.8 Hz, 1H), 7.54 (ddd, J = 8.8, 2.1, 0.7 Hz, 1H), 3.38 (s, 3H), 2.62 (s, 3H). ^13^C NMR (101 MHz, CDCl_3_) δ 161.59, 159.98, 158.21, 138.25, 135.64, 132.32, 124.59, 120.21, 118.29, 114.11, 112.90, 103.49, 77.43, 77.32, 77.11, 76.80, 26.64, 16.74, 0.09. HRMS (FAB-MS), *m*/*z*): calcd. for C14H10Br2N4O2, 426.07; found: 425.91 [M]+. FT-IR (cm^−1^): 2964 (OH), 2221 (C≡N), 1744 (C=O), 1702–1262 (C=C),1584 (N=N), 1232 (C-N), 1035 (C-Br), 926–500 (C-H).

### 2.3. Fabrication and Measurement of the Dye-Based Color Filters

Color-resistant solutions were prepared using the synthesized azo pyridone derivatives and other components. The other components of the solutions included an acrylic binder consisting of methyl methacrylate groups, carboxylic acid groups, and benzyl methacrylate groups, a leveling agent, and PGMEA as a solvent. The solution was coated onto a 2.5 cm × 2.5 cm transparent glass substrate using a MIDAS SPIN-1200D spin-coater at 850 rpm for 10 s. All of the fabricated color filters were post-baked at 220 °C for 3 min. The ΔE_ab_ values were measured using a multichannel spectrophotometer (Otsuka MCPD-3000) before and after the post-baking treatment.

## 3. Results and Discussion

For the molecular design of new yellow dyes with good optical and thermal properties, we selected the azo pyridone group as a core moiety with excellent optical properties. The synthesis routes are described in Figure 2. The azo pyridone group can be generally synthesized in three steps. To ensure adequate solubility, we attached an alkyl group as well as halogen groups (i.e., fluoride, chloride, and bromide). The final compounds were denoted as M-PAMOPC, F-PAMOPC, Cl-PAMOPC, and Br-PAMOPC, respectively, which were synthesized by Sandmeyer, diazolation, and coupling reactions. All of the compounds were purified by recrystallization and subsequently characterized using NMR spectroscopy (see Section 2). Relevant information regarding the characterization of the synthesized compounds is listed in Appendix A.

For a colorant to be used as a color-filter material for camera image sensors, it should be soluble in PGMEA solvent, which is commonly used in the display and camera industry. Thus, as previously mentioned, we attached a halogen group (i.e., F, Cl, or Br) onto an azo pyridone core moiety. The solubility data are summarized in Table 1. The solubility value is obtained when the solution maintains a transparent color at maximum solute concentration. The four dyes are soluble in PGMEA, and F-PAMOPC and Cl-PAMOPC in particular show greater than 0.1 wt% solubility. F-PAMOPC exhibited the highest solubility of 0.5 wt%, resulting in a wide addition amount window. It can be explained by the relatively strong electronegativity property of fluoride atom.

The optical data are presented in Figure 1 and Table 2. Transmittance spectra were recorded for the dyes at a concentration of 1 × 10^−4^ M in PGMEA, which corresponds to real application conditions used commercially. The requirement for the yellow colorant millbase is a transmittance of less than 5% at 435 nm and greater than 90% at 530 nm. As shown in Table 2, the four synthesized materials and Y138-millbase (commercial pigment) satisfy the requirements for commercial use. This requirement for the yellow colorant arises from the requirement for high color purity as well as that for low color interference in a small image-sensor pixel, and this yellow material can be added to a green millbase colorant to provide high color purity through blue-light absorption for a high-resolution image-sensor pixel. Compared with the band-edge value of 490 nm for Y138-millbase, the three synthesized compounds (F-PAMOPC, Cl-PAMOPC, and Br-PAMOPC) exhibit similar band-edge values of 493, 497, and 498 nm, respectively. The maximum absorption wavelength values are similar, and the transmittance of the four compounds is less than 0.16% at 435 nm and greater than 90% at 530 nm. These results indicate that these compounds can be incorporated into the green color material for commercial image sensors. In the case of the region between the maximum absorption and band edges, all the compounds exhibit a steep slope similar to that of the Y138-millbase; this spectral shape is related to color purity.

To investigate the thermal properties and solvent resistance properties of the four dyes and, thus, their potential for commercialization, we measured their color difference, ΔE_ab_, before and after the thermal stability and solvent resistance tests (Table 3). Solutions of the four compounds were spin coated, and the resultant films were pre-baked for 10 min at 100 °C. Each sample was exposed to 365 nm light with an intensity of 400 mJ/cm^2^ and was post-baked for 3 min at 220 °C. For the thermal properties, the sample was also baked for 10 min at 220 °C. For the solvent resistance test, the sample was immersed for 10 min into PGMEA solvent and then dried, and the color difference of ΔE_ab_ was subsequently determined. High thermal and solvent resistance properties, as indicated by a ΔE_ab_ value of less than 5, are critical for dyes used in the commercial colorant process [10,11,16,17]. In the case of thermal properties, Cl-PAMOPC and Br-PAMOPC showed stable ΔE_ab_ values of 4.93 and 3.43, indicating that these dyes can be used in commercial applications. However, F-PAMOPC exhibited a ΔE_ab_ value of 6.61, which is close to but does not satisfy the requirement for commercial use. If the mixture ratio of the colorant and other additives is varied and optimized, the thermal properties might be improved. A comparison of the thermal resistance of the halogen derivatives and the methyl derivative reveals that the halogen group increased the thermal resistance because of the high electronegativity of the halogen groups. However, among the three compounds with different halogen atoms, another effect was the increase in thermal resistance with increasing atomic mass of the different halogen atoms from fluoride to bromide. As a result, Br-PAMOPC exhibited the highest thermal resistance among the three halogenated derivatives. This result is consistent with the solvent resistance results as well as the TGA experimental data. In the case of solvent resistance, Cl-PAMOPC (ΔE_ab_ = 3.88) and Br-PAMOPC (ΔE_ab_ = 3.63) satisfy the commercial requirement of a ΔE_ab_ value of less than 5 (Table 3), and F-PAMOPC (ΔE_ab_ = 6.53) is close to meeting the requirement. J. Choi group reported a similar thermal stability experiment result with their compounds. Thermal treatment condition was 250 °C and 60 min, and the ΔE_ab_ value was in the range of 2.1~28.1. Although it also has high thermal stability data, it is not soluble in PGMEA commercial solvent and has limitations in the case of dye application [15]. The comparison is difficult since no papers use the exact same experimental conditions, but our result shows high performance compared with the conventional yellow material. By increasing atomic mass from fluoride to bromide, the solvent resistance improved. Thus, according to the optical and thermal data among the investigated materials, Cl-PAMOPC is the best-suited material for use as a yellow colorant additive for image sensors.

To evaluate the thermal properties of the synthesized materials, the degradation temperature (T_d_) corresponding to 5% weight loss was measured via TGA. Figure 2 and Table 4 show the TGA traces and data, respectively. All the compounds showed high thermal stability, with a T_d_ greater than 250 °C. By increasing the atomic number of the halogen group of the halogenated compounds, the T_d_ increased from 261 to 286 °C, which can be explained by the effects of the increasing molecular weight and increasing electronegativity. For comparison, M-PAMOPC, which has a methyl group, exhibited a T_d_ of 266 °C.

We prepared thin films of the four synthesized materials and subsequently evaluated their surface properties using AFM (Figure 3). The M-PAMOPC sample exhibited a root mean square (RMS) roughness of 57 nm, and the three halogenated compounds showed RMS roughness values ranging from 24 to 30 nm. When a halogen was incorporated into the chemical structure, the film surface became smooth. Halogenation can thus provide a relatively dense film and enable film formation via an easy solution processing method.

## 4. Conclusions

Novel azo yellow dye compounds were successfully synthesized as image-sensor colorant additives. Four materials were proposed and evaluated on the basis of their optical, thermal resistance, chemical resistance, and surface properties. By optimizing the effects of the electronegativity and molecular weight of different halides, we optimized the performance of the colorants. Among the four investigated compounds, F-PAMOPC and Cl-PAMOPC showed superior solubility in PGMEA (greater than 0.1 wt%), and all three halogenated compounds exhibited optical properties similar to those of the commercialized yellow pigment Y138-millbase in terms of the band edge and maximum absorption wavelengths. A comparison of the related performance of the four compounds revealed that Cl-PAMOPC exhibited optimum optical and thermal properties, including a transmittance of less than 5% at 435 nm and greater than 90% transmittance at 530 nm, a ΔE_ab_ of less than 5 both after a thermal treatment and after a solvent resistance test, and a T_d_ greater than 250 °C. The results indicate that Cl-PAMOPC can be used as an image-sensor colorant.

## Data Availability

Not applicable.

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
