# Peer review of "Novel Yellow Azo Pyridone Derivatives with Different Halide Atoms for Image-Sensor Color Filters"

_molecules, 2022, doi:10.3390/molecules27196601_

Round 1

Reviewer 1 Report

In work by Park et al., four new dyes derived from azo pyridone with different halogens were synthesized from commercially available dihalogenated anilines and their eventual application as image-sensor color filters. I enjoyed reading the manuscript. However, there are some flaws that I would like to have reviewed by the authors and some information added to the paper before its publication in Molecules journal.

Major points:

1) The introduction is concise. This is good because the reader can retain in his mind the essential information regarding the topic that is covered in the manuscript. The problem the authors desire to address is very well identified. However, this one is very poor in bibliographic references. The literature poorly supports the sentences. I advise keeping the structure but adding new appropriate references that can refer the author to works within the scope of the work.

2) The manuscript's greatest weakness is the characterization of new compounds which is not complete. (i) FTIR spectra were not acquired, and bands are not described in the characterization of compounds. In my view, this is an essential technique for identifying compounds since not all research teams have NMR devices at their disposal. (ii) Melting points of the compounds should also be determined. The 1H spectra show that the compounds were obtained with a high degree of purity, but these techniques are essential for a complete characterization of the new compounds! If they weren't new, 1H NMR would be enough to show that they've successfully synthesized them. (iii) The coupling constants of the main 1H signals should be indicated. (iv) 13C NMR should also have been performed. Why didn't you get them? Didn't have enough? (v) Mass spectrometry to confirm the mass of the dyes?

In these previous points, I ask that you at least be able to determine the melting points and obtain the FTIR spectra. And, please, justify in the manuscript why the other methodologies were not carried out for the complete characterization of novel compounds.

3) Reaction yields are not indicated. The yields of all the synthesis steps must be shown in Scheme 2. Thus, the author cannot understand the ease with which the prepared dyes were obtained.

4) The experimental section should include how the solubility of the compounds in PGMEA was determined. In addition, the results and discussion should elucidate a brief explanation of how the introduction of different halogens resulted in such different solubility in PGMEA.

6) The different properties tested are well described (thermal resistance, chemical resistance and surface properties), and the results are well interpreted based on the results shown in the tables and figures. However, the results obtained by the authors with some works described in the literature are not discussed. This is another major weakness of the present work. The conclusions section should also include a sentence related to that comparison. It is crucial to understand if, compared to other studies, the compounds presented here have improved properties.

Minor comments:

1) Lines 61, 89, 100, 110, 122: The number 1 of 1H must be superscript

2) Lines 89, 100, 101, 122: The number 3 of CDCl3 must be subscript

3) Lines 210, 212, ... : Td (degradation temperature) is not always identified in the same way; to standardize.

4) Table 4: How were the molar weights of the compounds obtained? Have they merely calculated values? Explain.

5) Figure 3: Aesthetically not very appealing. I would advise reducing the size of the letters that identify the various panels (a), (b), (c) and (d). For a better analysis, the size of the digits on the axes should be increased.

Having addressed the vast majority of my comments, I would be delighted to accept publication of your manuscript for publication.

Reviewer 2 Report

This article described the synthesis, the optical and thermal properties of azo pyridone derivatives. These dyes are described as potential candidates for replacing the Y138 yellow dye (commercial dye).

The paper is clear and easy to understand. However, I strongly recommend the authors to reconsider the publication as a communication. The content is not sufficient (analysis and discussion) for a full paper. Furthermore, the authors claim that they design new dyes. Yet three of the four dyes are already described in old patents (Japanese, Chinese and Korean patents). Thus only F-PAMOPC can be claimed as a novelty. Similarly, 7 of the 12 references refer to the same researcher; I'm not an expert on this branch of dye engineering, however I doubt that there are no other relevant and more recent publications on this subject. Thus, I ask the authors to greatly improve their references with more recent and diverse ones.

Regarding the contents of the article I have several questions and remarks :

1) Obtained mass, yields, final product aspect (colour, solid or liquid phase) and 13C NMR are missing for all reported syntheses, as well as 1H NMR for the first synthesis (2.2.1). The synthesis of the dyes M/F/Br/Cl-PAMOPC follows the same procedure with minor adaptations, thus I ask the authors to write a general procedure.

2) On table 3, the authors carry out thermal and solvent resistance tests comparing the different dyes. To me, similar tests on the Y138 dye are missing and should be provided.

3) Spectral data for colour stress tests are missing and this hinder the understanding of thermal effect. The authors emphasize that slight colour change is due to the increases of the molecular weight. Keeping this in mind, I wonder if the authors have try to get information from thermal stability from DSC measurements ? Is-it possible that colour changes are due to local phase modifications ?

4) Figures 1 and 2, please adapt the legend to make it understandable for black & white printing and add Y-138 dye on the label of the figure 1.

5) General remark for the NMR figures (S1-S4), it’s very difficult to see the different peaks as well as their shape and multiplicity. Can the authors explain me the peaks at -4.86 and -4.92 ppm for S3 and S4 respectively? The 1H NMR of Br-MPAMOPC clearly evidence a large amount of ethanol in the sample.

Round 2

Reviewer 1 Report

I appreciated the great effort made by the authors to improve the manuscript significantly.

Final comments:

1) The intro has been improved significantly. In addition, several bibliographic references were added to guide the reader to the bibliography, most of which are very recent.

2) The main shortcoming of the manuscript was the characterization of the compounds, which was significantly improved. FTIR spectra, melting points, proton coupling constants, and all signals relating to 13C were added to the manuscript. The spectra at 13C show, once again, the high purity of all synthesized compounds. Mass spectrometry was also performed, and the authors improved all the points I suggested.

3) Reaction yields were also added. However, it is not possible to understand if this income is global or exclusive to the last reaction step. I ask that you please clarify this.

4) As requested, it was added to the manuscript as the authors determined the solubility in PGMEA.

5) Regarding the discussion, I think it remains little explored. Despite this, the authors explain that the comparison is difficult since no papers use the same experimental conditions. I don't think you have to use the same experimental conditions for discussion. Each group uses its methodology, and not all protocols are standardized, and this should not be decisive for the non-comparison of results.

6) The authors have reviewed all my minor comments. However, regarding table 4, it does not seem plausible to use the values from the Chemdraw program. If you did mass spectrometry, why do you use those values? I think they should be changed or even removed from that same table.

Thus, in my view, the manuscript can be accepted for publication in the Molecules journal after minor revisions.

Reviewer 2 Report

I thank the authors for improving the manuscript, however there are still some minors points:

* For the synthesis 2.2.1, 2.2.2, 2.2.3 and 2.2.4, weights of the obtained products are missing

* in mat. & instrumen. section (2.1), you mentioned that HPLC was performed but in the manuscript there are no evidence that HPLC was used for experiments.

* I'm still not comfortable with the use of the word "designed" (page 2, l47) since only one compounds is a new design. This could be misleading for readers to me.

Regarding the great improvement of the article, it can be published under minor revision (see points above)
